# *Helicobacter pylori* Infection and Chronic Immune Thrombocytopenia

**DOI:** 10.3390/jcm11164822

**Published:** 2022-08-17

**Authors:** Hiroaki Takeuchi, Aoi Okamoto

**Affiliations:** Department of Medical Laboratory Sciences, Health and Sciences, International University of Health and Welfare Graduate School, 4-3 Kouzunomori, Narita-city 286-8686, Japan

**Keywords:** *Helicobacter pylori* infection, immune thrombocytopenia (ITP), platelet activation/aggregation, pathophysiological etiologies and pathways, strain diversity, flexible *H. pylori* community (*H. pylori* flora)

## Abstract

Approximately half of the world’s population is infected with *Helicobacter pylori*, which causes gastric disease. Recent systematic reviews and meta-analyses have reported that *H. pylori* may also have extragastric manifestations such as hematologic diseases, including chronic immune thrombocytopenia (cITP). However, the molecular mechanisms by which *H. pylori* induces cITP remain unclear, and may involve the host immune response, bacterial strain diversity, and delivery of bacterial molecules to the host blood vessels. This review discusses the important pathophysiological mechanisms by which *H. pylori* potentially contributes to the development of cITP in infected patients.

## 1. Introduction

*Helicobacter pylori* is a gram-negative spiral pathogenic bacterium that mainly causes upper gastrointestinal diseases such as inflammation, hyperplastic polyp, ulcer, and cancer [1,2]. Approximately half of the world’s population is estimated to have been infected with *H. pylori* at some point; typically, the infection occurs in childhood, and persistent, lifelong infection of the stomach can occur if left untreated [3]. Epidemiological research has documented that bacterial virulence factors differ between Western and East Asian strains of *H. pylori*, and the East Asian strains are thought to be more highly pathogenic [4]. Pathogenic *H. pylori* is highly genetically diverse and adapts to survive in the microenvironments/conditions present in different anatomical parts of individual patients’ stomachs [5]. Furthermore, *H. pylori* bacteriophages (phages) also promote the development of a flexible *H. pylori* community with variable characteristics [6]. Thus, the *H. pylori* population within the stomach is continuously changing in response to changes in conditions, which contributes to persistent infection and the development of considerable biological polymorphism [5].

According to many reports, *H. pylori* also causes extragastric disease, including blood diseases such as iron deficiency anemia, vitamin B_12_ deficiency, and chronic immune thrombocytopenia (cITP), as well as other diseases such as metabolic syndrome, diabetes, non-alcoholic fatty liver disease, Alzheimer’s disease, neurologic disease, skin disease, and *cardiovascular disease* [7,8,9,10]. The cardiovascular diseases caused by *H. pylori*, including atherosclerosis and acute coronary syndrome, develop because of vessel damage and the appearance of plaques due to inflammatory cytokine production and platelet activation following infection [9,11]. Eradication therapy for *H. pylori* infection has been covered by insurance since 2013 in Japan, which was the first country in the world to do this [12], and since then the insurance coverage has been expanded to include treatment of *H. pylori*-induced extragastric diseases such as cITP. This review focuses on the relationship between *H. pylori* infection and cITP. Relevant papers were retrieved by searching the MEDLINE/PubMed database using the terms “*Helicobacter*”, “platelet”, and “ITP/thrombocytopenia.”

## 2. Immune Thrombocytopenia and *H. pylori* Infection

Immune thrombocytopenia (ITP), an acquired autoimmune disease, is characterized by the presence of antiplatelet antibodies that lead to platelet destruction and a reduced platelet count (thrombocytopenia, <100 × 10^9^/L). ITP can be a primary disease or can develop as a secondary etiology, such as infection with microorganisms including *H. pylori* and others [13,14]. Thrombocytopenia that is continuously observed for at least 12 months is referred to as chronic ITP (cITP). cITP is typically treated with immunosuppressant therapy, and splenectomy can be performed if medical treatment is unsuccessful. However, how antiplatelet antibodies are produced and whether the antibodies trigger the development of cITP remain unclear. Gasbarrini et al. first reported that *H. pylori* eradication therapy improved the platelet count in *H. pylori*-positive cITP patients (*H. pylori*-associated cITP), suggesting a pathophysiological relationship between *H. pylori* infection and cITP [15]. Since then, studies have shown that eradication therapy partially or completely restores the platelet count in more than 50% of *H. pylori*-infected patients with cITP, with some geographical/regional variability [16,17,18]. Especially, the rate of effective response to eradication therapy on cITP is approximately more than 80% in Middle East region/Mediterranean [19]. In *H. pylori*-infected cITP patients, the effective response to eradication therapy is relatively high in Middle East, Asia, and Europe, and low in North America. A study involving North American cITP patients demonstrated that *H. pylori*-infected Hispanic cITP patients responded more effectively to *H. pylori* eradication than did white, non-Hispanic cITP patients. Current guidelines in multiple countries recommend *H. pylori* eradication in *H. pylori*-infected cITP patients who are unresponsive to traditional treatment for cITP. However, more pathophysiological research is needed to elucidate the etiology and pathophysiological mechanisms by which *H. pylori* infection influences the development of cITP.

## 3. Mechanistic Pathway of ITP Development

### 3.1. Molecular Mimicry and Cross-Reaction

Human immuno-complexity, which involves the interactions among infectious agents, environmental and hormonal factors, inflammation, and the host immune response, is well known to trigger autoimmunity by different pathways. Previous reports have shown that various autoimmune disorders are most likely associated with *H. pylori*, including cITP [8,20,21,22,23]. The mechanistic pathways by which *H. pylori* promotes cITP development are postulated to include molecular mimicry, increased plasmacytoid dendritic cell count, and the host immune response to virulence factors such as vacuolating-associated cytotoxin gene A (VacA) and cytotoxin-associated gene A (CagA). Michel et al. performed antibody profiling with platelets from three *H. pylori*-infected ITP patients and found no *H. pylori*-specific antibodies. However, they did find autoantibodies to platelet surface glycoproteins (IIb/IIIa or Ib) that did not directly react with *H. pylori* molecules such as CagA, VacA, UreB, HspA, FsB, FlaA, and UreA [24]. Since then, additional studies have shown that the molecular mimicry between *H. pylori*-derived molecules as CagA and VacA, and platelet surface glycoproteins (IIb/IIIa or Ib) is responsible for cITP induction [25,26,27,28,29]. The anti-CagA and/or anti-VacA antibodies react to platelets (cross-reaction), leading to platelet aggregation and destruction [26,28,29]. These platelet-associated IgG (PAIgG) antibodies derived from *H. pylori* infection are predictive of platelet recovery after *H. pylori* eradication, suggesting that antibody titers could be used as a marker to determine the effectiveness of *H. pylori* eradication in *H. pylori*-infected cITP patients [27]. In particular, *H. pylori* CagA was identified an effector molecule that translocates into host gastric epithelial cells via a type IV secretion system, and has also been discovered in exosomes isolated from blood samples [30], indicating that it can directly affect organs (tissues and cells), including platelets. Future studies should focus on bacterial components encapsulated not only in exosomes but also in outer membrane vesicles (OMVs) released from bacteria to clarify the pathophysiological mechanisms of *H. pylori*-associated disorders including cITP.

The evidence for cross-reaction based on molecular mimicry described above is not sufficient to exactly explain the pathophysiological pathway by which *H. pylori* causes cITP. Studies have shown that the titers of antibodies such as anti-CagA antibodies in *H. pylori*-associated cITP patients who are responsive to eradication therapy (responders) decrease, but do not reach the negative level, within the few weeks following therapy. In addition, decreased antibody titers are frequently observed even in non-responders. Interestingly, in our cases and in previous reports, an increased platelet count is observed in responders in the first few weeks following eradication therapy [31]. In addition, even in responders with unsuccessful eradication therapy, the platelet count temporarily increases and then returns to its original level, most likely because of numbers of bacteria grown again in the stomach. These clinical manifestations are difficult to explain based on antibody cross-reaction pathways. Thus, the platelet response should be monitored on a weekly basis following eradication therapy to quantitatively determine temporal fluctuations in platelet count and antigen and antibody levels.

Two recent studies proposed an alternative pathway for cITP induction by *H. pylori* in which an immunocomplex forms comprising the *H. pylori* antigen Lpp20, *H. pylori*-specific antibodies, and platelets [32,33]. These studies demonstrated that the *H. pylori*-specific antibodies (anti-Lpp20 antibodies) were present at significantly higher levels in responders compared with in non-responders and in *H. pylori*-uninfected cITP patients. The anti-Lpp20 antibodies reacted to *H. pylori* Lpp20 but not platelets; however, Lpp20 can bind to platelets, resulting in immunocomplex formation and platelet destruction. This alternative pathway could be responsible for the clinical manifestations mentioned above, as *H. pylori*-specific platelet-binding antigens could decrease quickly and temporally even in responders with unsuccessful eradication therapy. Lpp20 is released from *H. pylori* and may have high antigenicity [34,35]. It is typically contained in membranous vesicles or masked by the bacterial outer membrane, so anti-Lpp20 antibodies are not always produced in *H. pylori*-infected patients [36]. However, significant levels of anti-Lpp20 antibodies are detected in patients with *H. pylori*-associated cITP, implying that Lpp20 triggers an immune response in certain *H. pylori*-infected patients. This indicates that an individual’s immune response to *H. pylori* Lpp20 may be involved in the development of *H. pylori*-associated cITP. Thus, future studies should focus on genetic and strain diversities, as well as the variation in *H. pylori* communities within individual stomachs, and microparticles (exosomes and OMVs) should be analyzed to elucidate the pathophysiological etiologies of *H. pylori*-associated cITP (Figure 1).

### 3.2. Platelet Activation and Aggregation

Lpp20 directly binds to platelets, eventually leading to platelet activation and aggregation. Some *H. pylori* strains induce platelet activation and aggregation, resulting in platelet destruction [23,31,37]. The platelet activation/aggregation pathway is thought to involve interactions between anti-*H. pylori* IgG and IgG receptors (FcgRIIA) on platelets, as well as between von Willebrand factor (vWf) on *H. pylori* and platelet surface glycoproteins Ib/IX (gp-Ib/IX (CD42)). Consequently, the interaction between vWf and gp-Ib causes activation of gp-IIb/IIIa (CD41/CD61) and irreversible binding of vWf to platelets [37]. The occurrence of this type of *H. pylori*-induced platelet aggregation in the gastric microvasculature and/or blood vessel may lead to the development of systemic disorders such as cardiovascular diseases. Previous reports have demonstrated a correlation between *H. pylori* infection and coronary artery diseases including atherosclerosis, dyslipidemia, and thrombosis [8,22,38]. Platelet aggregation prompts the formation of atherosclerotic lesions, which is accompanied by multiple events such as plaque formation, inflammatory cytokine release, damaged endothelial cells, fibrous cap formation, and thrombosis [9,38,39]. *H. pylori*-specific DNA was observed in atherosclerotic plaques in severe cardiovascular disease patients [40], and *H. pylori* CagA encapsulated in exosomes was discovered in blood samples [30]. In addition, *H. pylori* infection is associated with increased levels of the exosome-associated miRNA miR-25, even in peripheral blood, and this miRNA is known to regulate the nuclear factor (NF)-κB signaling pathway [41]. Taken together, these findings indicate that *H. pylori* infection may be involved in the development of cardiovascular diseases and propose a pathway by which delivery of as yet unidentified *H. pylori* molecules to the blood vessels trigger the induction of these diseases though platelet aggregation and dysfunction, endothelial damage, and other processes [41].

Interestingly, the mechanical pathway described, based on platelet activation and aggregation, does not involve CagA or VacA [42]. So far, the exact *H. pylori* molecules that are responsible for platelet activation and aggregation have not been identified. Lpp20 was recently discovered to directly bind to platelets and eventually lead to platelet activation and aggregation, suggesting that it may promote thrombosis in *H. pylori*-infected patients with the absence of anti-Lpp20 antibodies. More research is needed to evaluate whether and how this molecule triggers and/or enhances the induction of *H. pylori*-associated disorders.

### 3.3. Host Immune Response to H. pylori Infection

Another potential mechanism for *H. pylori*-associated cITP is inhibition of Fcγ receptors on monocytes, which eventually results in increased antiplatelet antibodies and platelet turnover by reducing FcγRIIB levels. Reduced FcγRIIB production and the presence of autoreactive B cells increase the phagocytic activity of monocytes, leading to decreased platelet counts [43,44]. In addition, the number of plasmacytoid dendritic cells, a type of antigen-presenting cells, was found to be increased in patients with *H. pylori*-associated cITP [45,46]. In gastric epithelium infected with *H. pylori*, dendritic cells enhance host immune response with an influx of T cells and induce IL-12 and IL-10 release. Furthermore, *H. pylori* outer membrane proteins (Omp) such as HpaA and Omp 18 proteins stimulate the production of IL-12 and IL-10 by dendritic cells [46,47]. In addition to CagA and VacA, other *H. pylori* molecules such as outer inflammatory protein A (OipA), blood group antigen-binding adhesion A (BabA), sialic acid-binding adhesion (SabA), and picB have also been reported to be involved in *H. pylori*-associated disorders based on the host immune response to *H. pylori* infection [23,48]. Tyrosine phosphorylation of the Glu-Pro-Ile-Tyr-Ala (EPIYA) motif in CagA is a strong antigenic factor that induces IL-8- and NF-κB-mediated inflammation, as well as the production of anti-CagA antibodies, which cross-react with platelets, as described above [49]. VacA suppresses helper T-cells via interruption of the T-cell receptor IL-2 pathway [50]. VacA also binds to multimerin-1 and elastin microfibril interfacer 4 on megakaryocytes and platelets, respectively, and enhances platelet activation and clearance [28]. Taken together, these findings suggest that these host immune response pathways play a role in the pathophysiological progression of *H. pylori*-associated cITP.

## 4. Conclusions

Even though the exact mechanism remains unclear, multiple reviews and meta-analyses regarding *H. pylori*-associated extragastric disorders, including cITP, have outlined the broad pathophysiological mechanisms by which *H. pylori* directly and/or indirectly contributes to the development of cITP. Guidelines from different countries around the world, including Japan, agree that cITP is an extragastric manifestation of *H. pylori* infection, and that eradication therapy is recommended [51,52,53,54]. In fact, many cases of platelet count recovery after eradication therapy (responders) have been reported in different countries. On the other hand, some patients do not experience platelet recovery after successful eradication therapy (non-responders). At present, little information is available regarding how to predict which patients will be responders and which will be non-responders. The *H. pylori* protein Lpp20 and anti-Lpp20 antibodies may be useful markers in this context. More research is needed to identify predictive markers by obtaining a more accurate understanding of the biological and immunological events that occur in the *H. pylori*-infected host. Furthermore, the clinical manifestations of cITP in *H. pylori*-infected patients, including platelet activation/aggregation, appear to be dependent on *H. pylori* strain variation. Thus, more attention should be paid to the diversity of *H. pylori* strains in individual patients’ stomachs, as well as ongoing changes in the composition of the *H. pylori* population in the stomach. Finally, *H. pylori* molecules have been detected encapsulated in serum-derived extracellular vesicles such as exosomes and OMVs, indicating that they can directly and functionally interact with multiple organs/tissues/cells outside the stomach. Thus, there are multiple promising avenues for future research regarding the pathophysiological etiologies of *H. pylori*-associated cITP.

## Figures and Tables

**Figure 1 jcm-11-04822-f001:**
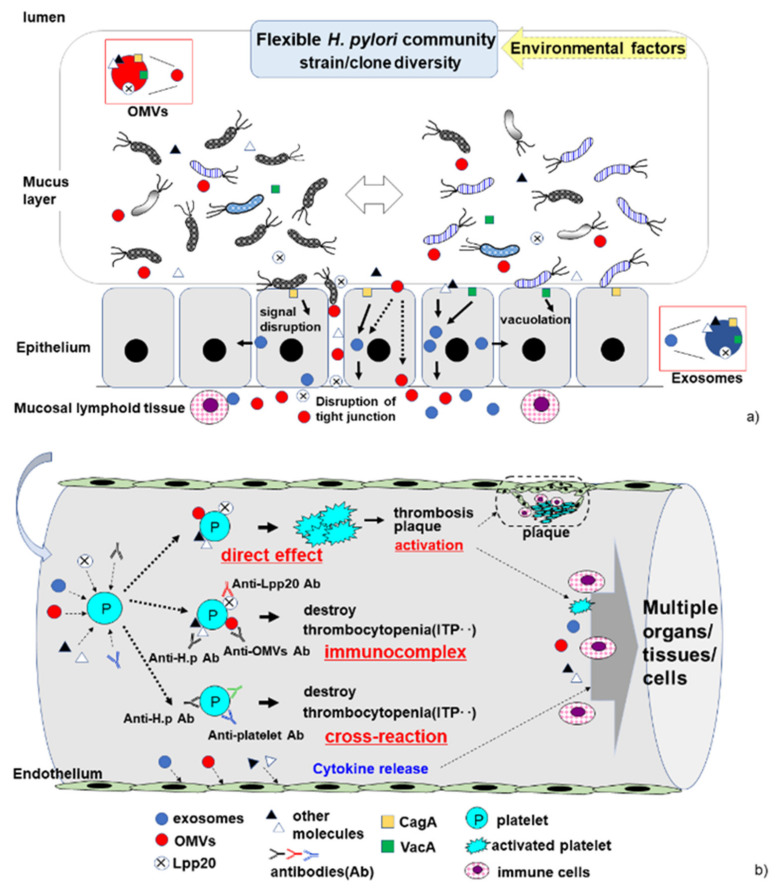
The continuously changing *H. pylori* community in the stomach, and proposed pathophysiological pathways by which *H. pylori* induces cITP. (**a**) In the stomach, the strain/clone composition of the *H. pylori* community is highly variable. The bacteria release molecules such as outer membrane vesicles (OMVs) containing pathogenic antigens (i.e., CagA and VacA). Bacterium–epithelial cell interactions can induce host cell damage, disrupt host signal transduction, and lead to the production of exosomes. The bacterial molecules and host cytokines trigger an immune response in mucosal lymphoid tissue. (**b**) In the vessel, the proposed pathophysiological pathways by which *H. pylori* induces cITP, including cross-reaction, immunocomplex formation, and direct interaction of *H. pylori* molecules with platelets in the host blood vessels. Exosomes, OMVs, and molecules released by *H. pylori* affect platelets and the endothelium in the presence or absence of anti-*H. pylori* antibodies. These bacterial factors (exosomes, OMVs, Lpp20, and other molecules) may bind directly to platelets, leading to platelet activation. The production of a variety of anti-*H. pylori* antibodies contributes to platelet destruction via cross-reaction and immunocomplex formation in *H. pylori*-associated cITP.

## Data Availability

Not applicable.

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
