# Peer review of "Helicobacter pylori Infection and Chronic Immune Thrombocytopenia"

_jcm, 2022, doi:10.3390/jcm11164822_

Round 1

Reviewer 1 Report

The presented review is contribute to a better understanding the important pathophysiological mechanisms by which H. pylori potentially participate to the development of chronic immune thrombocytopenia in infected patients. The authors concluded that the exact mechanism remains unclear. There are multiple reviews and meta-analyses regarding H. pylori-associated extragastric disorders. Guidelines from different countries around the world, including Japan, agree that chronic immune thrombocytopenia is an extragastric manifestation of H. pylori infection, and that eradication therapy is recommended. There will be an overall benefit to publishing this work. The material is very interesting, original and significant. The English language is appropriate and understandable.

Author Response

Dear Reviewer 1,

I appreciate for your review and comments. At present, many previous researches and our evidence show the positive relationship between H. pylori infection and development of cITP, however, the pathophysiological mechanisms remain unclear. In this review, I updated new potential mechanisms focused on the status of H. pylori colonization in the stomach (changing H. pylori community and microvesicles from the community) and new molecules (Lpp20) discovered which directly bind to platelets.

In near future, I believe that more evidence concerned with H. pylori and cITP must elucidate the pathophysiological mechanisms of H. pylori-associated cITP in detail. For that, we need to focus the true infection status of H. pylori in the stomach mentioned above and geographical difference on H. pylori strain (characterization).

Thank you so much again for your comments.

Best wishes,

Hiroaki Takeuchi

Reviewer 2 Report

Dear Authors,

Thank you for these very important details in the field of HP, however, I believe that we need more focus on the information regarding the "Immune Thrombocytopenia and H. pylori Infection" please try to find more references to enrich the paragraph from worldwide, especially in the pathophysiological relationship between H. pylori infection and cITP you need more details from other regions such as the Eastern Mediterranean region, as you know the type and the infected with Helicobacter pylori is different between the regions and there are some publications said the in EMR the infected of HP more than 80%, moreover, I would like to congratulate you for the valuable review 

Best regards

Author Response

Dear Reviewer 2,

I appreciate for your review and comments. I understand and agree with your comments regarding to geographical difference on H. pylori strains (characterization).

I cited the reference [18] concerned with international survey including Middle East region on H. pylori test and cITP (standardization committee) and added “Especially, the rate of effective response to eradication therapy on cITP is approximately more than 80% in Middle East region/Mediterranean [19]“ into the position (line 63, page 2).

Except for mechanisms reviewed in this paper, to my knowledge, there is no new potential mechanism to understand the clinical manifestation so far. However, I believe to obtain new insight through more investigations with characterizing the aspect of H. pylori strains (Mediterranean) whose origin seems to be mixed with Europe and Asia strains. This is very interesting.

In this review, I updated new potential mechanisms focused on the status of H. pylori colonization in the stomach (changing H. pylori community and microvesicles from the community) and new molecules (Lpp20) discovered which directly bind to platelets.

In near future, I believe that more evidence concerned with H. pylori and cITP must elucidate the pathophysiological mechanisms of H. pylori-associated cITP in detail. For that, we need to focus the true infection status of H. pylori in the stomach mentioned above and geographical difference on H. pylori strain (characterization) as you know.

I appreciate if you would understand and agree to our thinking.

Your comments are very helpful for our future study.

Thank you so much again.

Best wishes,

Hiroaki Takeuchi